# Do rapid BMI growth in childhood and early-onset obesity offer cardiometabolic protection to obese adults in mid-life? Analysis of a longitudinal cohort study of Danish men

Laura D Howe,[1] Esther Zimmermann,[2] Ram Weiss,[3] Thorkild I A Sørensen[2,4]

▶ Prepublication history and additional material for this paper is available. To view please visit the journal (http://dx.doi.org/10.1136/bmjopen-2014-004827).

For numbered affiliations see end of article.

**Correspondence to**
Dr Laura D Howe;
laura.howe@bristol.ac.uk

## ABSTRACT

**Objective:** Some obese individuals have no cardiometabolic abnormalities; they are 'metabolically healthy, but obese' (MHO). Similarly, some non-obese individuals have cardiometabolic abnormalities, that is, 'metabolically at risk, normal weight' (MANW). Previous studies have suggested that early-onset obesity may be associated with MHO. We aimed to assess whether body mass index (BMI) in childhood and early-onset obesity are associated with MHO.

**Setting:** General population longitudinal cohort study, Denmark.

**Participants:** From 362 200 young men (mean age 20) examined for Danish national service between 1943 and 1977, all obese men (BMI ≥31 kg/m$^2$, N=1930) were identified along with a random 1% sample of the others (N=3601). Our analysis includes 2392 of these men attending a research clinic in mid-life (mean age 42). For 613 of these men, data on childhood BMI are available. We summarised childhood BMI growth (7–13 years) using a multilevel model. Early-onset obesity was defined as obesity at examination for national service.

**Outcome measurement:** We defined metabolic health at the mid-life clinic as non-fasting serum cholesterol <6.6 mmol/L, non-fasting glucose <8.39 mmol/L and pulse pressure <48 mm Hg. Participants were categorised into four groups according to their obesity (BMI ≥30 kg/m$^2$) and metabolic health in mid-life.

**Results:** 297 of 1097 (27.1%) of obese men were metabolically healthy; 826 of 1295 (63.8%) non-obese men had at least one metabolic abnormality. There was no evidence that rapid BMI growth in childhood or early-onset obesity was associated with either MHO or the MANW phenotype, for example, among obese men in mid-life, the OR for MHO comparing early-onset obesity with non-early-onset obesity was 0.97 (95% CI 0.85 to 1.10).

**Conclusions:** We found no robust evidence that early-onset obesity or rapid BMI growth in childhood is protective for cardiometabolic health.

### Strengths and limitations of this study

- This is the largest study until now to examine the question of whether childhood body mass index (BMI) and early-onset obesity offer cardiometabolic protection to people who are obese as adults.
- Using non-fasting measures of cholesterol and glucose and pulse pressure, we categorised men in mid-life as either metabolically healthy, but obese (MHO); metabolically abnormal, obese; metabolically healthy, normal weight or metabolically abnormal, normal weight, and examined the relationships of childhood BMI growth and early-onset obesity with this phenotype.
- Our measures of cholesterol and glucose were from non-fasting blood samples. There is therefore potential misclassification in our measure of 'metabolically healthy'. We have tried to address this difficulty by using thresholds defined by percentiles of our data, selecting a relatively high cut-off for glucose in order to have reasonable specificity and sensitivity. Given that our estimated prevalence of MHO is within the range seen in previous studies, we feel that this is unlikely to have strongly biased our results.
- We found no consistent evidence that rapid BMI growth in childhood, or early-onset obesity, is associated with a favourable cardiometabolic profile, contradicting previous smaller studies that have examined this question.
- Our findings imply that the notion of greater severity and metabolic impact of weight gain during adulthood is doubtful and that prevention measures should be implemented in the early childhood years.

## BACKGROUND

Recent studies, including reviews,[1–3] have drawn attention to the phenotype 'metabolically healthy, but obese' (MHO).[4] [5] MHO individuals are defined as those who are

obese but have a 'healthy' cardiometabolic profile; precise definitions have varied between studies, but tend to include one or more of insulin resistance, lipids, blood pressure and markers of inflammation. The prevalence of MHO is estimated to be between 10% and 40% in obese people (with the remainder classified as metabolically at risk, obese (MAO)).[1 6–9] Likewise, non-obese populations can be classified as either 'metabolically healthy, normal weight' (MHNW) or 'metabolically at risk, normal weight' (MANW).

Some studies have shown that MHO is associated with a reduced risk of mortality, cardiovascular disease events, type II diabetes or a lower carotid intima-media thickness compared with MAO.[10–14] However, other studies, including a recent systematic review,[15] have demonstrated similar or greater rates of mortality,[15] or development of hypertension[16–18] and type II diabetes[17 18] in MHO compared with MAO or MHNW individuals, or high rates of conversion of MHO individuals to an MAO state,[19 20] suggesting that MHO is an intermediate step in the development of MAO rather than a static condition, and that no level of obesity is healthy.[15 21 22]

Several factors have been shown to be positively associated with MHO, for example, female gender, higher physical activity (although not in all studies[23]), greater fitness,[24] longer sleep duration[23] and a favourable psychosocial profile.[1] There is also some evidence that childhood growth is associated with MHO; one small study (N=117 obese children, mean age 10.4 years) found higher birth weight and faster weight gain in infancy was associated with greater levels of circulating insulin and insulin resistance, whereas weight gain after the age of 4 years was negatively associated with insulin resistance.[25] Analyses of 499 obese adults in Helsinki, Finland, found that those who had the metabolic syndrome were similar in birth size compared with obese people who did not have metabolic syndrome, but were lighter and thinner between 2 and 11 years of age.[26] A further study of 43 postmenopausal women observed that an earlier age of onset of obesity is more common in MHO, with 48% of MHO women reporting an age of obesity onset <20 years compared with 29% of MAO women.[27] These findings are intriguing, given the well-established association between elevated body mass index (BMI) in childhood and greater risk of coronary heart disease in adulthood.[28 29] Research examining the influence of growth in childhood or early-onset obesity and MHO is, however, rare, since few studies have detailed data on childhood growth and cardiometabolic health in adulthood.

In this paper, we take a cohort of men from Denmark, for whom data are available on childhood growth, early-onset obesity and cardiometabolic risk factors measured in mid-life. We examine the associations of early-onset obesity and trajectories of BMI growth in childhood (from 7 to 13 years) with MHO, MAO, MHNW and MANW.

## MATERIALS AND METHODS
### Participants, exposure measurements and sample definition
National service is compulsory in Denmark, and before entry into service, draftees attend a physical examination where their fitness to serve is assessed. In the 1970s, a study was initiated to make use of the anthropometric data collected at these examinations. From the 362 200 men examined in the metropolitan area of Copenhagen between 1943 and 1977, all 1930 men who were obese (defined as 35% overweight relative to a Scandinavian standard in use at the time, equivalent to a BMI ≥31 kg/m²) and a random sample of 1% of the others (N=3601) were selected to form a cohort study. Only 3% of men eligible for national service were not examined for medical reasons, with obesity not being a condition that allowed exemption.[30] A further 2% of men were not examined because they had volunteered for service before the age of 18 years. Height was measured without shoes, and weight in only underwear. The mean age at examination was 20 years, range 18–31.

Two re-examinations were undertaken of this cohort, first, in 1982–1984 (between 4 and 40 years after the draft board examination, mean age 35 years, range 22–64) and second, in 1991–1994 (mean age of participants 45.6 years, range 33–75). All of the cohort members who were obese at draft board examination and (for logistical reasons) half of the controls were invited to these clinics, which were attended by 2075 and 1703 participants, respectively. Previous analyses of this cohort have shown that lower BMI and higher intelligence, educational level, social class and age were associated with a greater probability of attending follow-up clinics.[31] Further details of this cohort are published elsewhere.[30 32–35] As part of a separate study,[36 37] anthropometric data extracted from school health records between ages 7 and 13 are available for a subset of individuals. Participants gave informed consent, and the protocol was consistent with the principles of the Declaration of Helsinki.

For our main analysis (association between early-onset obesity and obesity-metabolic status in mid-life), we included all participants who attended either the first or second follow-up clinic and had their BMI, systolic and diastolic blood pressure (SBP/DBP), cholesterol and glucose measured. We used data from the second follow-up clinic where available, or the first follow-up clinic for participants attending the first but not second assessment. For our secondary analysis of the association between childhood growth trajectories and obesity-metabolic status in mid-life, this sample was further restricted to participants with one or more BMI measure from school health records between 7 and 13 years.

### Exposure measurements
Obesity status (BMI ≥31 kg/m²) in early adulthood (at entry into national service, mean age 20 years) determines our measurement of early-onset obesity.

We used a multilevel model to estimate BMI at age 7 and the linear change in BMI between 7 and 13 years for all participants with one or more BMI measure. Multilevel models are an appropriate tool for the analysis of longitudinal data, since they allow for the non-independence of repeated measures on the same individual, and can estimate a full trajectory for all participants with one or more measurement under a missing at random assumption.[38 39] The multilevel model was run in MLwiN V.2.26[40] via Stata V.12,[41] using the command runmlwin.[42] Full details of the model are provided in online supplementary material. From the multilevel model, we derived predicted BMI at age 7 and predicted linear change in BMI between 7 and 13 years; these variables were used as exposures in our analyses.

## Outcome measurements

At each follow-up clinic, height was measured without shoes and weight in only light clothes. SBP and DBP were measured three times with the participant at rest; the mean of the three measures was used. Total serum glucose and cholesterol were assessed from non-fasting blood samples (examinations were undertaken during daytime, and the participants were not requested to skip breakfast or lunch) drawn from a vein after light stasis. Samples were placed in a box containing ice water, and within 2 h centrifuged at 3000 rpm for 10 min. The plasma was stored at 4°C and analysed the following day. Cholesterol was measured enzymatically using the CHOD-PAP method at the first examination and high-performance CHOD-PAP at the second (Boehringer Mannheim). Glucose was assayed using the hexokinase/G6P-DH assay. Blood samples remained at room temperature for 20–60 min, after which the whole blood from the EDTA tube was placed in a plastic tube and frozen at −20°C. To control for treatment effects, a constant of 10 mm Hg was added to SBP and 5 mm Hg to DBP for individuals reporting treatment with antihypertensive medication (n=156); these values are consistent with the effects of antihypertensive medications, and this method has been shown to be a reasonable approach for reducing treatment bias and has been used in previous studies.[43–45] Pulse pressure, identified in previous research as an important independent risk factor for cardiovascular disease,[46] was defined as SBP–DBP.

For each cardiometabolic risk factor (SBP, DBP, pulse pressure, glucose and cholesterol), a measure of the level of the risk factor given what would be predicted from age and BMI was created by regressing the risk factor on age and BMI at the time of assessment, and standardising the resulting residuals to have a mean of 0 and variance of 1. Thus for these measures a value of zero would mean that the level of the risk factor is what would be predicted given age and BMI, positive values would mean higher than expected levels, and vice versa for negative values.

Various definitions of MHO have been used in previous studies.[1 2] Since the blood samples in our study were non-fasting, standard cut-offs based on fasting samples would be too low. We therefore opted to define cut-offs for cholesterol and glucose based on percentiles of the study-population distribution. For cholesterol, we deemed levels above the 75th centile (6.6 mmol/L) to be high. Fluctuations in glucose are greater than those in cholesterol, so a higher threshold is necessary in order to obtain a measure that is reasonably sensitive but also has sufficient specificity. Therefore, we defined high glucose levels as above the 95th centile for non-obese participants (8.39 mmol/L). In line with previous research,[46] we defined high pulse pressure as above 48. 'Metabolically healthy' was defined as having cholesterol, glucose and pulse pressure levels below these thresholds; participants with one or more measurement above these thresholds were deemed 'metabolically at risk'. Individuals were classified according to their obesity status in mid-life (obesity defined as ≥30 kg/m$^2$) and metabolic status, such that individuals were MHO, MAO, MHNW or MANW. As sensitivity analyses, we repeated our analyses using lower cut-points for glucose (≤5.6 mmol/L) and cholesterol (<5.2 mmol/L), and using SBP and DBP instead of pulse pressure.

## Other data

The occupation of participants' fathers was self-reported in questionnaires and coded as manual or non-manual. A binary indicator of participants' smoking behaviour (any vs no smoking) was created from self-reported data.

## Statistical analysis

The associations of early-onset obesity and childhood growth with the cardiometabolic risk factors and standardised residuals from regression of cardiometabolic risk factors on age and BMI were assessed using linear regression. The associations of early-onset obesity and childhood growth with MHO (compared with MAO) or with MHNW (compared with MANW) were assessed using logistic regression. As a sensitivity analysis, we assessed whether our findings are robust to adjustment for participants' fathers' social class and participants' smoking status.

## RESULTS

In total, 2392 participants were included in our analyses; of these, data from the second follow-up assessment were available for 1691 and from the first assessment for a further 701 (table 1).

## Prevalence of obesity and metabolic phenotypes

Among the 2392 participants, 1295 were not obese. Of these, 469 (36.2%) were classified as MHNW, and 826 (63.8%) as MANW (table 1). Thus 63.8% of normal weight individuals had one or more metabolic abnormality. Of the 1097 participants who were obese at follow-up, 297 (27.1%) were MHO.

**Table 1** Sample characteristics, mean (SD) or percentage of characteristics of participants included in analyses

| | Participants included in analysis of the association between early-onset obesity and metabolic health in mid-life | | | Participants included in analysis of the association between childhood growth and metabolic health in mid-life | | |
|---|---|---|---|---|---|---|
| | Participants attending second follow-up in adulthood N=1691 | Participants attending first follow-up in adulthood but not the second follow-up N=701 | Combined whole sample N=2392 | Participants attending second follow-up in adulthood N=420 | Participants attending first follow-up in adulthood but not the second follow-up N=193 | Combined whole sample N=613 |
| Age (years) | 45.61 (7.93) | 34.81 (8.38) | 42.44 (9.45) | 48.01 (6.82) | 36.34 (6.90) | 44.34 (8.73) |
| Number (%) obese in early adulthood* | 784 (46.4%) | 346 (49.4%) | 1130 (47.2%) | 189 (45.0%) | 95 (49.2%) | 284 (46.3%) |
| Body mass index (kg/m$^2$) | 30.57 (6.69) | 29.76 (6.87) | 30.34 (6.75) | 30.52 (6.79) | 31.00 (7.74) | 30.67 (7.10) |
| Systolic blood pressure (mm Hg) | 142.37 (18.12) | 138.50 (16.18) | 141.89 (18.32) | 143.64 (18.75) | 138.08 (16.33) | 141.89 (18.19) |
| Diastolic blood pressure (mm Hg) | 92.81 (11.41) | 86.45 (12.25) | 91.27 (12.31) | 92.29 (10.81) | 89.27 (11.70) | 91.34 (11.18) |
| Pulse pressure (mm Hg) | 49.96 (13.26) | 52.22 (12.38) | 50.62 (13.04) | 51.36 (13.73) | 48.80 (11.91) | 50.55 (13.23) |
| Glucose (mmol/L) | 6.48 (2.95) | 6.57 (2.42) | 6.51 (2.80) | 6.56 (3.14) | 6.65 (2.52) | 6.59 (2.96) |
| Cholesterol (mmol/L) | 6.10 (1.28) | 5.25 (1.22) | 5.85 (1.32) | 6.11 (1.20) | 5.34 (1.24) | 6.86 (1.26) |
| Number (%) with metabolic phenotypes† | | | | | | |
| Non-obese at mid-life follow-up | N=903 | N=392 | N=1295 | N=235 | N=102 | N=337 |
| Metabolically healthy | 328 (36.3%) | 141 (36.0%) | 469 (36.2%) | 81 (34.5%) | 49 (48.0%) | 130 (38.6%) |
| Metabolically at risk | 575 (63.7%) | 251 (64.0%) | 826 (63.8%) | 154 (65.5%) | 53 (52.0%) | 207 (61.4%) |
| Obese at mid-life follow-up | N=788 | N=309 | N=1097 | N=185 | N=91 | N=276 |
| Metabolically healthy, but obese | 221 (28.0%) | 76 (24.6%) | 297 (27.1%) | 48 (25.9%) | 28 (30.8%) | 76 (27.5%) |
| Metabolically at risk, obese | 567 (72.0%) | 233 (75.4%) | 800 (72.9%) | 137 (74.1%) | 63 (69.2%) | 200 (72.5%) |

*Defined as 35% overweight relative to a Scandinavian standard in use at the time, equivalent to a body mass index (BMI) ≥31 kg/m$^2$.
†Metabolically healthy is defined by having pulse pressure (systolic–diastolic blood pressure) <48 mm Hg, glucose <8.39 mmol/L and cholesterol <6.6 mmol/L; that is, any one or more metabolic abnormality confers 'metabolically at-risk' status. Obesity at the mid-life follow-up is defined by BMI ≥30 kg/m$^2$.

Compared with MANW individuals, MHNW individuals tended to be younger (mean and SD for age 44.93 (10.89) and 42.09 (8.40) years, respectively) and thinner (mean and SD for BMI 25.55 (2.69) and 24.82 (2.85) years; table 2). MAO had higher BMI compared with MHO individuals (mean and SD for BMI 36.49 (5.10) and 35.79 (4.53) kg/m$^2$), but there was little difference in age (table 2).

## Childhood growth trajectories

Differences in BMI between participants who were obese and non-obese at draft board examination were already evident by age 7. The mean BMI at age 7 and rate of BMI change between 7 and 13 years were higher in those who were obese at draft board (see online supplementary table S1). The multilevel model had good fit to the data (see online supplementary table S2 and figures S1–S3).

## Associations of childhood growth and early-onset obesity with cardiometabolic health and obesity/metabolic phenotypes in mid-life

Those with early-onset obesity (ie, those who were obese at draft board examination) had a BMI in mid-life on average 9.65 kg/m$^2$ (95% CI 9.27 to 10.03) greater than those who were not obese at the draft board examination. Greater BMI at age 7, faster rate of BMI increase between 7 and 13 years and early-onset obesity were all associated with greater SBP, DBP and pulse pressure and greater glucose levels in mid-life (table 3). These associations tended to increase in magnitude after adjustment for age at the research clinic and year of draft board examination. In our analyses, the associations of childhood growth and early-onset obesity with levels of cholesterol were weaker than for the other cardiometabolic risk factors.

There was very little evidence of associations of childhood growth and early-onset obesity with standardised scores of the cardiometabolic risk factors that account for age and BMI at the time of assessment (table 4). There was no evidence of associations for the standardised scores of SBP or DBP. For standardised scores of glucose, early-onset obesity and faster rate of BMI increase between 7 and 13 years were associated with higher scores (no association was seen for BMI at 7 years). For example, after adjustment for year at draft board, early-onset obesity was associated with a 0.16 SD higher standardised score of glucose (95% CI 0.08 to 0.24), and a 1 kg/m$^2$/year increase in the rate of BMI growth between 7 and 13 years (equivalent to approximately 2 SD of the rate of growth) was associated with a 0.26 SD higher standardised score (95% CI 0.03 to 0.49). Conversely, early-onset obesity, greater BMI at age 7 and faster rate of BMI increase between 7 and 13 years were all associated with lower standardised scores of cholesterol. For example, after adjustment for year of draft board examination, early-onset obesity was associated with a −0.19 lower standardised score for cholesterol (95% CI −0.27 to −0.11) and a 1 kg/m$^2$/year increase in the rate of BMI growth between 7 and 13 years was associated with a −0.12 lower standardised score (95% CI −0.35 to 0.10). Thus early-onset obesity and faster rates of BMI increase in childhood are associated with a higher glucose level but lower cholesterol level than would be predicted from age and adult BMI.

Among those who were not obese at the follow-up clinic, early-onset obesity was associated with slightly lower odds of being MHNW compared with MANW, but the CI was wide and included the null value (OR 0.76, 95% CI 0.54 to 1.07; table 5). There was no consistent relationship between childhood BMI trajectories and MHNW compared with MANW.

Among those who were obese in mid-life, there was a weak association between early-onset obesity and lower odds of being MHO compared with MAO; OR 0.69, 95% CI 0.46 to 1.04. There was no association between

**Table 2** Characteristics of participants within each metabolic and obesity category

| | MHNW* | MANW* | MHO* | MAO* |
|---|---|---|---|---|
| *Mean (SD)* | | | | |
| | N=469 | N=826 | N=297 | N=800 |
| Age (years) | 42.09 (8.40) | 44.93 (10.89) | 40.15 (7.33) | 40.93 (8.53) |
| BMI (kg/m$^2$) | 24.82 (2.85) | 25.55 (2.69) | 35.79 (4.53) | 36.49 (5.10) |
| Systolic blood pressure (mm Hg) | 126.60 (10.37) | 142.97 (16.98) | 134.70 (10.74) | 152.41 (18.22) |
| Diastolic blood pressure (mm Hg) | 86.85 (9.21) | 88.00 (12.34) | 95.21 (9.77) | 95.77 (12.76) |
| Pulse pressure (mm Hg) | 39.74 (5.23) | 54.97 (11.97) | 39.49 (5.67) | 56.64 (12.73) |
| Cholesterol (mmol/L) | 5.26 (0.87) | 6.06 (1.40) | 5.39 (0.82) | 6.15 (1.45) |
| Glucose (mmol/L) | 5.62 (0.87) | 6.29 (2.30) | 5.76 (0.74) | 7.53 (3.95) |
| Number (%) that were obese in early adulthood† | 61 (13.0) | 118 (14.3) | 253 (85.2) | 698 (87.3) |

*Metabolically healthy is defined by having pulse pressure (systolic–diastolic blood pressure) <48 mm Hg, glucose <8.39 mmol/L and cholesterol <6.6 mmol/L; that is, any one or more metabolic abnormality confers 'metabolically at-risk' status. Obesity in mid-life is defined by BMI ≥30 kg/m$^2$.
†Obesity in early adulthood is defined as 35% overweight relative to a Scandinavian standard in use at the time, equivalent to a BMI ≥31 kg/m$^2$.
BMI, body mass index; MANW, metabolically at risk, normal weight; MAO, metabolically at risk, obese; MHNW, metabolically healthy, normal weight; MHO, metabolically healthy, but obese.

**Table 3** Linear regression coefficients (with 95% CIs) for the association of early-onset obesity and childhood body mass index (BMI) trajectories with cardiovascular risk factors measured in mid-life

| Model* | Exposure | Outcome | | | | | |
|---|---|---|---|---|---|---|---|
| | | BMI (kg/m$^2$) | Systolic blood pressure (mm Hg) | Diastolic blood pressure (mm Hg) | Pulse pressure (mm Hg) | Glucose (mmol/L) | Cholesterol (mmol/L) |
| *Full sample, N=2329* | | | | | | | |
| 1 | Early-onset obesity† | 9.65 (9.27 to 10.03) p<0.001 | 7.70 (6.27 to 9.14) p<0.001 | 5.31 (4.35 to 6.28) p<0.001 | 2.39 (1.35 to 3.44) p<0.001 | 1.14 (0.92 to 1.36) p<0.001 | −0.14 (−0.25 to −0.04) p=0.01 |
| 2 | | 9.76 (9.36 to 10.16) p<0.001 | 10.28 (8.82 to 11.74) p<0.001 | 6.02 (5.01 to 7.02) p<0.001 | 4.27 (3.21 to 5.33) p<0.001 | 1.30 (1.07 to 1.53) p<0.001 | −0.07 (−0.18 to 0.04) p=0.19 |
| *Subsample with childhood growth data, N=613* | | | | | | | |
| 1 | BMI at 7 years (kg/m$^2$) | 1.81 (1.60 to 2.02) p<0.001 | 1.37 (0.72 to 2.02) p<0.001 | 0.72 (0.32 to 1.12) p<0.001 | 0.65 (0.17 to 1.13) p=0.01 | 0.12 (0.02 to 0.23) p=0.03 | −0.04 (−0.09 to 0.00) p=0.08 |
| 2 | | 1.75 (1.53 to 1.97) p<0.001 | 1.82 (1.17 to 2.47) p<0.001 | 0.85 (0.44 to 1.26) p<0.001 | 0.97 (0.49 to 1.45) p<0.001 | 0.13 (0.02 to 0.24) p=0.02 | −0.02 (−0.06 to 0.03) p=0.50 |
| 1 | BMI change 7–13 years (kg/m$^2$/) | 10.95 (10.01 to 11.89) p<0.001 | 8.47 (5.26 to 11.67) p<0.001 | 4.53 (2.55 to 6.51) p<0.001 | 3.93 (1.57 to 6.30) p=0.001 | 1.31 (0.78 to 1.83) p<0.001 | −0.20 (−0.42 to 0.03) p=0.09 |
| 2 | | 8.75 (7.59 to 9.91) p<0.001 | 9.69 (5.76 to 13.63) p<0.001 | 4.76 (2.27 to 7.25) p<0.001 | 4.93 (2.00 to 7.87) p=0.001 | 1.58 (0.91 to 2.24) p<0.001 | 0.05 (−0.22 to 0.33) p=0.71 |

*Model details: model 1 is unadjusted; model 2 is adjusted for age at outcome assessment and year of draft board examination, and the coefficient for change in BMI between 7 and 11 years is additionally adjusted for BMI at 7 years as predicted by the multilevel model.
†Defined as 35% overweight relative to a Scandinavian standard in use at the time, equivalent to a BMI ≥31 kg/m$^2$ in early adulthood, mean age 20 years.

**Table 4** Linear regression coefficients (with 95% CIs) for the association of early-onset obesity and childhood BMI trajectories with levels of cardiovascular risk factors in mid-life adjusted for age and BMI in mid-life

| Model* | Exposure | Outcomes† (SD, given age and concurrent BMI) | | | | |
|---|---|---|---|---|---|---|
| | | SBP | DBP | Pulse pressure | Glucose | Cholesterol |
| *Full sample, N=2329* | | | | | | |
| 1 | Early-onset obesity‡ | 0.02 (−0.06 to 0.10) p=0.62 | −0.01 (−0.09 to 0.07) p=0.78 | 0.04 (−0.04 to 0.12) p=0.38 | 0.13 (0.05 to 0.21) p=0.002 | −0.13 (−0.21 to −0.05) p=0.002 |
| 2 | | 0.05 (−0.04 to 0.13) p=0.28 | −0.06 (−0.14 to 0.03) p=0.18 | 0.11 (0.03 to 0.19) p=0.01 | 0.16 (0.08 to 0.24) p<0.001 | −0.19 (−0.27 to −0.11) p<0.001 |
| *Subsample with childhood growth data, N=613* | | | | | | |
| 1 | BMI at 7 years (kg/m$^2$) | 0.00 (−0.04 to 0.04) p=0.97 | −0.01 (−0.05 to 0.03) p=0.57 | 0.01 (−0.03 to 0.05) p=0.61 | −0.01 (−0.05 to 0.02) p=0.42 | −0.03 (−0.07 to 0.00) p=0.07 |
| 2 | | 0.00 (−0.03 to 0.04) p=0.89 | −0.01 (−0.05 to 0.02) p=0.47 | 0.01 (−0.02 to 0.05) p=0.49 | −0.01 (−0.05 to 0.02) p=0.50 | −0.04 (−0.08 to −0.01) p=0.02 |
| 1 | BMI change 7–13 years (kg/m$^2$/year) | 0.03 (−0.15 to 0.21) p=0.75 | −0.04 (−0.22 to 0.14) p=0.69 | 0.07 (−0.11 to 0.25) p=0.46 | 0.11 (−0.07 to 0.29) p=0.24 | −0.15 (−0.33 to 0.03) p=0.10 |
| 2 | | 0.05 (−0.18 to 0.28) p=0.65 | −0.02 (−0.25 to 0.20) p=0.83 | 0.07 (−0.16 to 0.31) p=0.52 | 0.26 (0.03 to 0.49) p=0.03 | −0.12 (−0.35 to 0.10) p=0.28 |

The outcome in these analyses is a measure of the level of the cardiovascular risk factor (SBP, DBP, pulse pressure, glucose and cholesterol) given what would be predicted from age and BMI; this was created by regressing the risk factor on age and BMI at the time of assessment, and standardising the resulting residuals to have a mean of 0 and variance of 1. Thus for these measures a value of zero would mean that the level of the risk factor is what would be predicted given age and BMI, positive values would mean higher than expected levels, and vice versa for negative values.

*Model details: model 1 is unadjusted; model 2 is adjusted for year of draft board examination, in model 2 the coefficient for change in BMI between 7 and 11 years is additionally adjusted for BMI at 7 years as predicted by the multilevel model.

†Cardiovascular risk factors were regressed on concurrent BMI and age, and standardised residuals (mean of 0, variance of 1) from those regressions are used as the outcomes in these analyses. Thus a positive value of these outcomes means that an individual has a higher level of that outcome than would be expected given their age and BMI, and a negative value means the opposite, lower level of the outcome given their age and BMI.

‡Defined as 35% overweight relative to a Scandinavian standard in use at the time, equivalent to a BMI ≥31 kg/m$^2$ in early adulthood, mean age 20 years.

BMI, body mass index; DBP, diastolic blood pressure; SBP, systolic blood pressure.

**Table 5**  ORs (with 95% CIs) for the association of obese status in early adulthood and growth in early childhood with metabolic status in mid-life; stratified by mid-life obesity status

| Model* | Metabolically healthy, normal weight† | Metabolically at risk, normal weight† | Metabolically healthy, but obese† | Metabolically at risk, obese† |
|---|---|---|---|---|
| *Full sample, N=2329* | | | | |
| | N=469 | N=826 | N=297 | N=800 |
| Early-onset obesity‡ | | | | |
| 1 | 0.90 (0.64 to 1.25) p=0.52 | 1 (ref) | 0.84 (0.57 to 1.23) p=0.37 | 1 (ref) |
| 2 | 0.76 (0.54 to 1.07) p=0.12 | 1 (ref) | 0.69 (0.46 to 1.04) p=0.08 | 1 (ref) |
| *Subsample with childhood growth data, N=613* | | | | |
| | N=130 | N=207 | N=76 | N=200 |
| BMI at 7 (kg/m$^2$) | | | | |
| 1 | 0.97 (0.85 to 1.10) p=0.60 | 1 (ref) | 0.99 (0.88 to 1.13) p=0.92 | 1 (ref) |
| 2 | 0.93 (0.81 to 1.07) p=0.32 | 1 (ref) | 0.97 (0.85 to 1.10) p=0.65 | 1 (ref) |
| BMI change 7–13 (kg/m$^2$/year) | | | | |
| 1 | 0.85 (0.41 to 1.70) p=0.62 | 1 (ref) | 1.50 (0.73 to 3.05) p=0.27 | 1 (ref) |
| 2 | 0.66 (0.25 to 1.72) p=0.39 | 1 (ref) | 1.40 (0.61 to 3.48) p=0.38 | 1 (ref) |

Coefficients are ORs (95% CIs) comparing obese in early adulthood with non-obese in early adulthood, or a one unit increase in body mass index (BMI) at age 7 or the linear rate of change in BMI between 7 and 13 years.
*Model details: model 1 is unadjusted; model 2 is adjusted for age at outcome assessment and year of draft board examination. In model 2 the coefficient for change in BMI between 7 and 13 years is additionally adjusted for BMI at 7 years as predicted by the multilevel model.
†Metabolically healthy is defined by having pulse pressure (systolic–diastolic blood pressure) <48 mm Hg, glucose <8.39 mmol/L and cholesterol <6.6 mmol/L; that is, any one or more metabolic abnormality confers 'metabolically at-risk' status. Obesity in mid-life is defined by BMI ≥30 kg/m$^2$.
‡Defined as 35% overweight relative to a Scandinavian standard in use at the time, equivalent to a BMI ≥31 kg/m$^2$ in early adulthood, mean age 20 years.

BMI at age 7 or BMI increases between 7 and 13 years and MHO (table 5). Results were unchanged when adjusted for participants' fathers' social class and participants' smoking status (see online supplementary table S3).

Associations of early-onset obesity and child growth with MHO remained similar if lower thresholds for defining abnormal cholesterol and glucose were used, or if SBP and DBP were used in the definition of metabolically healthy instead of pulse pressure (results available from authors on request).

## DISCUSSION

We found little consistent evidence to support the hypothesis that rapid BMI growth between 7 and 13 years and early-onset obesity are associated with a protective cardiometabolic effect. Early-onset obesity and faster rates of BMI increase in childhood were associated with a higher glucose level but lower cholesterol level than would be predicted from age and adult BMI. No associations were seen for SBP or DBP levels or pulse pressure adjusted for concurrent age and BMI. There was also no consistent evidence that early-onset obesity or rates of BMI change in childhood were associated with being 'metabolically healthy'—early-onset obesity

was weakly associated with lower odds of being MHO, but no associations were seen for BMI at age 7 or rate of BMI change between 7 and 13 years. Our findings contrast with a study of 43 postmenopausal women, which observed that 48% of MHO women reported an age of obesity onset <20 years compared with 29% of MAO women.[27] The earliest available childhood BMI measures in our cohort are from age 7; thus it is possible that BMI changes at earlier stages could be associated with being metabolically healthy, and we are thus unable to make direct comparisons between our results and those of the studies looking at birth size and weight gain in early childhood in relation to MHO.[25][26] Our findings are supported by some previous research on this cohort, which has found that the men who were obese at their draft board examination in early adulthood experience a doubling of all-cause mortality risk,[47] as well as increased risks of mortality from cardiovascular diseases and a wide range of other specific causes.[48] However, other analyses of this cohort have demonstrated that obese men, maintaining their weight since the draft board examination in early adulthood, had lower risk of impaired glucose tolerance than non-obese men in early adulthood who became similarly obese by age 51.[49]

In this cohort of Danish men, 27.1% of participants who were obese at follow-up were classified as MHO

defined by having non-fasting cholesterol <6.6 mmol/L, non-fasting glucose <8.39 mmol/L and pulse pressure <48 mm Hg. This is within the range of 10–40% prevalence reported in previous studies.[1 6–9] A large study of over 5000 adults in the National Health and Nutrition Examination Surveys (NHANES) from the USA found a prevalence of MHO of 51.3%.[50] However, this study differed from ours in two important ways: first, by considering overweight participants rather than obese, and second, by considering metabolically at risk to be two or more abnormalities, whereas we assumed one or more abnormality conferred risk.

Notably, a very high proportion of non-obese participants were classified as MANW: 63.8%. Analysis of data from a Chinese cohort also found a high prevalence of cardiometabolic abnormalities in non-obese adults,[51] leading the authors to conclude that focusing cardiovascular screening only on obese individuals would miss a large proportion of the at-risk population. The NHANES data also found that 23.5% of non-overweight participants had at least two metabolic abnormalities.[50] In our study, the mean BMI in the non-obese group was high—24.82 $kg/m^2$ in the MHNW group and 25.55 in the MANW group. Thus although not obese, many of these participants were overweight.

Our analysis included 2392 individuals, 1097 of whom were obese, making it larger than most existing studies on MHO.[1 2] We were only able to include a subset of these individuals (N=613) in our analysis of the association between childhood BMI growth and later metabolic health. This is a larger sample size than the former studies on this topic,[25 27] but we may still have had limited power to detect an association. We also were unable to examine the influence of growth before age 7 or of puberty. Furthermore, some of our participants were quite young at the follow-up examinations, and so associations may emerge in later life as cardiovascular risk develops. Our measure of early-onset obesity is based on obesity at a mean age of 20 years; while we feel that this is an appropriate definition of early-onset obesity in this population, which grew up at a time when childhood obesity was very rare, studies in younger cohorts may wish to consider earlier ages to define 'early' obesity. Our measures of adiposity are based on BMI, which is an imperfect measure of fatness, particularly in children. However, even in childhood, BMI has been shown to have similar associations with cardiovascular risk factors as directly determined fat mass.[52]

We did not have data on insulin or on lipids other than total cholesterol, or other measures that may be stronger predictors of cardiovascular events, and therefore we could not incorporate these into our definitions of MHO, MAO, MHNW and MANW. Our measures of cholesterol and glucose were from non-fasting blood samples. There is therefore potential misclassification in our measure of 'metabolically healthy'. We have tried to address this difficulty by using thresholds defined by percentiles of our data. Glucose levels are likely to fluctuate

more in non-fasting samples than cholesterol levels,[53] and we therefore selected a relatively high cut-off for glucose in order to have reasonable specificity and sensitivity. Given that our estimated prevalence of MHO is within the range seen in previous studies, we feel that this is unlikely to have strongly biased our results, which remained similar if lower cut-points were used for cholesterol and glucose.

We found no consistent evidence that rapid BMI growth in childhood, or early-onset obesity, is associated with a favourable cardiometabolic profile, contradicting previous smaller studies that have examined this question. Our findings imply that the notion of greater severity and metabolic impact of weight gain during adulthood is doubtful and that prevention measures should be implemented in the early childhood years.

## Author affiliations
[1]MRC Integrative Epidemiology Unit at the University of Bristol, School of Social and Community Medicine, University of Bristol, Bristol, UK
[2]Institute of Preventive Medicine, Frederiksberg and Bispebjerg Hospitals, The Capital Region, Copenhagen, Denmark
[3]Braun School of Public Health, The Hebrew University of Jerusalem, Jerusalem, Israel
[4]Faculty of Health and Medical Sciences, Novo Nordisk Center for Basic Metabolic Research, University of Copenhagen, Copenhagen, Denmark

**Contributors** The study was conceived by LDH and TIAS. The study was designed by LDH with input from TIAS and RW. LDH carried out statistical analyses and wrote the first draft of the manuscript. All authors contributed to critical revisions of the manuscript and approved the final submitted version.

**Funding** This work was supported by a UK Medical Research Fellowship to LDH (G1002375).

**Competing interests** None.

**Ethics approval** This study was conducted before the widespread introduction of ethical committees in Denmark; however, it was conducted according to the principles of the Helsinki Declaration.

**Provenance and peer review** Not commissioned; externally peer reviewed.

**Data sharing statement** Statistical code can be requested from the corresponding author. For details of data access and collaboration policy, please contact TIAS at tias@sund.ku.dk

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
