## [Reviewer comments · BMJ Open]

Some articles will have been accepted based in part or entirely on reviews undertaken for other BMJ Group journals. These will be reproduced where possible.

ARTICLE DETAILS

TITLE (PROVISIONAL)	Do rapid BMI growth in childhood and early-onset obesity offer cardiometabolic protection to obese adults in mid-life? Analysis of a prospective cohort study of Danish men
AUTHORS	Howe, Laura; Zimmermann, Esther; Weiss, Ram; Sorensen, Thorkild

VERSION 1 - REVIEW

REVIEWER	Johan Eriksson University of Helsinki Finland
REVIEW RETURNED	01-Mar-2014

GENERAL COMMENTS	The paper reviewed entitled “Do rapid BMI growth in childhood and early-onset obesity offer cardio-metabolic protection to obese adults in mid-life? Analyses of a prospective cohort study of Danish men” is focusing upon an interesting – and potentially important – research question. In general it is well written and easy to follow, the manuscript follows a logical order and the discussion is critical enough. There are some point that I feel should be raised: 1. The concept of “metabolically healthy obesity” has recently been challenged in a systematic review and meta-analysis published in 2013. This study should be referred to and discussed (Kramer CK, Zinman B, Retnakaran R. Are metabolically healthy overweight and obesity benign conditions? A systematic review and meta-analysis. Ann Intern Med 2013;159:758-69) and the accompanying editorial (Hill JO, Wyatt HR. The myth of healthy obesity. Ann Intern Med 2013;159:789)2. The definition of metabolic health is difficult. Since cholesterol concentration is not largely affected by the fasting/non-fasting state, a lower cut-off point would be more appropriate. The non-fasting glucose cut-point seems rather arbitrary as well as high. Did the authors consider/use other cut-off values and if yes how did this influence the findings. Why did the authors use pulse pressure instead on conventional blood pressure levels? The justification in the paper is not convincing and most scores and risk assessment tools use blood pressure values. Further the authors should discuss in more detail whether these are the most appropriate predictors of later cardiovascular health. Some information on duration of fasting should be provided.3. The authors report that among the normal weight individuals 63.8 % had at least one metabolic abnormality and were therefore classified as “abnormal”. Is this finding in line with findings from
--

other studies? I do not feel that a comparison to a Chinese study (ref 46) is appropriate since it is well known that metabolic disturbances occur in many Asian populations at lower BMIs and waist circumferences. Would it not be more appropriate to use as a criterion for metabolic abnormality the presence of at least two of the criterion used?

4. Page 8 : the authors state that "... the associations of childhood growthwith cholesterol ...". This is the case but, I feel this sentence needs to be rephrased and cholesterol is certainly a stronger risk factor for cardiovascular disease than fasting glucose. Perhaps the findings in relation to cholesterol should get some more attention in the paper.

5. First sentence in the discussion needs to be more specific. What the authors did find was that ..."rapid BMI growth from 7 to 13 years" – speaking about BMI growth in childhood is not appropriate since they focus upon only the age interval 7-13 years. Further previous studies have shown that most of the positive effects of early BMI growth is associated with BMI increase in early childhood or infancy – in other words much before the time period that is being focused upon in the present study.

6. Page 10, lines 12-15 – they are almost identical with the text in the introduction – please rephrase.

7. The paper examines the associations between obesity and BMI growth in childhood from 7 to 13 years. Puberty is included in this interval. Have the authors taken timing of puberty into considerations? How will puberty potentially influence the findings?

8. It could be argued that the authors do not study early obesity when focusing upon obesity around age 20 years. In relation to childhood growth again, the authors are most probably missing the age at BMI rebound and obesity beginning before 7 years of age. Would this affect the findings?

9. These study questions have been focused upon in the Helsinki Birth Cohort Study – e.g. Salonen MK, Kajantie E, Osmond C, Forsén T, Ylihärsilä H, Paile-Hyvärinen M, Barker DJ, Eriksson JG. Role of childhood growth on the risk of metabolic syndrome in obese men and women. *Diabetes Metab.* 2009 Apr;35(2):94-100 and Salonen MK, Kajantie E, Osmond C, Forsén T, Ylihärsilä H, Paile-Hyvärinen M, Barker DJ, Eriksson JG. Childhood growth and future risk of the metabolic syndrome in normal-weight men and women. *Diabetes Metab.* 2009 Apr;35(2):143-50 especially the first paper focuses upon issues very similar to the present study and should be discussed.

10. The age range being studied is extremely wide from 22 to 75 years, can this fully be taken into account by the statistical adjustments?

11. The problems associated with using BMI as a measure of obesity should be discussed in more detail.

12. How representative is the study population participating in the follow-up to the whole original population?

	Minor issues: 1. Is this a prospective study or would it be more appropriate to call it longitudinal 2. Page 2, l 37/38: mean – should be men
--	---

REVIEWER	Carrie Durward Utah State University, USA
REVIEW RETURNED	03-Mar-2014

GENERAL COMMENTS	In the abstract, line 38 "mean" should be "men". In some places (abstract, introduction) MHO is defined as "metabolically healthy, but obese", but other places (article summary, tables) it is "metabolically healthy, obese".
--

VERSION 1 – AUTHOR RESPONSE

Reviewer Name Johan Eriksson

The paper reviewed entitled “Do rapid BMI growth in childhood and early-onset obesity offer cardio-metabolic protection to obese adults in mid-life? Analyses of a prospective cohort study of Danish men” is focusing upon an interesting – and potentially important – research question. In general it is well written and easy to follow, the manuscript follows a logical order and the discussion is critical enough.

Response: We thank the review for their positive comments

There are some point that I feel should be raised:

1. The concept of “metabolically healthy obesity” has recently been challenged in a systematic review and meta-analysis published in 2013. This study should be referred to and discussed (Kramer CK, Zinman B, Retnakaran R. Are metabolically healthy overweight and obesity benign conditions? A systematic review and meta-analysis. *Ann Intern Med* 2013;159:758-69) and the accompanying editorial (Hill JO, Wyatt HR. The myth of healthy obesity. *Ann Intern Med* 2013;159:789)

Response: We thank the reviewer for bringing these references to our attention, and have now incorporated them into page 4 of the introduction as follows (newly added text underlined): “Some studies have shown that MHO is associated with a reduced risk of mortality, cardiovascular disease events, type II diabetes, or a lower carotid intima-media thickness compared with MAO.10-14 However, other studies, including a recent systematic review, have demonstrated similar or greater rates of mortality, or development of hypertension and type II diabetes in MHO compared with MAO or MHNW individuals, or high rates of conversion of MHO individuals to an MAO state, suggesting that MHO is an intermediate step in the development of MAO rather than a static condition, and that no level of obesity is healthy.”

2. The definition of metabolic health is difficult. Since cholesterol concentration is not largely affected by the fasting/non-fasting state, a lower cut-off point would be more appropriate. The non-fasting glucose cut-point seems rather arbitrary as well as high. Did the authors consider/use other cut-off values and if yes how did this influence the findings. Why did the authors use pulse pressure instead

on conventional blood pressure levels? The justification in the paper is not convincing and most scores and risk assessment tools use blood pressure values.

Response: We agree with the reviewer that deciding which cut-points to use to determine 'high' or 'unhealthy' levels of cholesterol or glucose is challenging because the measures were based on non-fasting samples. Our a priori decision was to use centiles of the distributions – defining high levels as above the 75th centile for cholesterol and above the 95th centile for glucose. We feel that this approach is the most justifiable given the non-fasting nature of the samples, and we selected a higher threshold for glucose compared with cholesterol due to the greater degree of fluctuation in glucose levels. We feel that using any lower cut-points would potentially result in classifying a very large number of individuals as 'unhealthy'. We did, however, carry out sensitivity analyses using lower cut-points - $<5.2\text{mmol/l}$ for cholesterol and $\leq 5.6\text{mmol/l}$ for glucose; the pattern of our results and conclusions remained the same. We also carried out sensitivity analyses to test the robustness of our conclusions to using systolic and diastolic blood pressure instead of pulse pressure, and again found that the results did not change. We have described these sensitivity analyses and the results of it in the manuscript as follows:

Methods (page 7): "As sensitivity analyses, we repeated our analyses using lower cut-points for glucose ($\leq 5.6\text{mmol/l}$) and cholesterol ($<5.2\text{mmol/l}$), and using systolic and diastolic blood pressure instead of pulse pressure."

Results (page 9): "Associations of early onset obesity and child growth with MHO remained similar if lower thresholds for defining abnormal cholesterol and glucose were used, or if systolic and diastolic blood pressure were used in the definition of metabolically healthy instead of pulse pressure (results available from authors on request)."

Discussion (page 11): ", which remained similar if lower cut-points were used for cholesterol and glucose"

Further the authors should discuss in more detail whether these are the most appropriate predictors of later cardiovascular health.

Response: Unfortunately we do not have data available on other predictors of cardiovascular risk. We have expanded the discussion of this on page 11 (newly added text underlined): "We did not have data on insulin or on lipids other than total cholesterol, or other measures that may be stronger predictors of cardiovascular events, and therefore we could not incorporate these into our definitions of MHO, MAO, MHNW and MANW."

Some information on duration of fasting should be provided.

Response: Examinations were undertaken during day time, and the participants were not requested to skip breakfast or lunch; we have now detailed this on page 6.

3. The authors report that among the normal weight individuals 63.8 % had at least one metabolic abnormality and were therefore classified as "abnormal". Is this finding in line with findings from other studies? I do not feel that a comparison to a Chinese study (ref 46) is appropriate since it is well known that metabolic disturbances occur in many Asian populations at lower BMIs and waist circumferences. Would it not be more appropriate to use as a criterion for metabolic abnormality the presence of at least two of the criterion used?

Response: The cited study of Chinese people uses a lower, Asia-appropriate cut off (BMI of 23) to define overweight, and so we feel that citing this study and comparing it with our findings is relevant. We also cited US data from analysis of NHANES and compared this with our findings. We have also discussed the multiple possible reasons underlying the high prevalence of metabolic abnormalities in

our non-obese participants, including the fact that the majority of men in this group were overweight.

4. Page 8 : the authors state that "... the associations of childhood growthwith cholesterol ...". This is the case but, I feel this sentence needs to be rephrased and cholesterol is certainly a stronger risk factor for cardiovascular disease than fasting glucose. Perhaps the findings in relation to cholesterol should get some more attention in the paper.

Response: We have rephrased this sentence to start with "in our study", but have retained the sentence since it is a true representation of our results to say that growth in childhood and early-onset obesity demonstrated far weaker associations with cholesterol levels than with the other cardiovascular risk factors we studied. Our analyses do not examine the comparative associations of cholesterol and other risk factors with cardiovascular events.

5. First sentence in the discussion needs to be more specific. What the authors did find was that ... "rapid BMI growth from 7 to 13 years" – speaking about BMI growth in childhood is not appropriate since they focus upon only the age interval 7-13 years. Further previous studies have shown that most of the positive effects of early BMI growth is associated with BMI increase in early childhood or infancy – in other words much before the time period that is being focused upon in the present study.

Response: We agree with the reviewer that we cannot generalise our findings to the whole of childhood. We have rephrased the sentence to read "between 7-13 years" instead of "in childhood". We have also added the following to page 11 to highlight our inability to study earlier childhood growth: "We also were unable to examine the influence of growth before age 7."

6. Page 10, lines 12-15 – they are almost identical with the text in the introduction – please rephrase.

Response: We have re-phrased this section so that it is different to the introduction.

7. The paper examines the associations between obesity and BMI growth in childhood from 7 to 13 years. Puberty is included in this interval. Have the authors taken timing of puberty into considerations? How will puberty potentially influence the findings?

Response: We have not considered the role of puberty in our analysis. Puberty is a potential confounder of our associations of interest, with early puberty possibly being associated with greater BMI, more rapid BMI increases, and more adverse levels of cardiometabolic risk factors. Although changes in height can in some circumstances be used to define a measure of pubertal development, with only annual measurements available, there is insufficient information to accurately determine pubertal development and such a measure would result in a considerable degree of misclassification in our participants. In addition, mean age at peak height velocity in males is at >13 years, so we would only be able to create a crude indicator of early versus non-early puberty. It is not known how early puberty might relate to the MHO phenotype, so it is not possible to speculate in which direction such confounding might have affected the results, although we would anticipate any effect being small. In some studies, controlling for pubertal development has not affected the observed associations between childhood growth and cardiometabolic risk (e.g. Howe LD et al. PLoS One 2010. e15186). We have added on page 11 the limitation that we were unable to control for pubertal development.

8. It could be argued that the authors do not study early obesity when focusing upon obesity around age 20 years. In relation to childhood growth again, the authors are most probably missing the age at BMI rebound and obesity beginning before 7 years of age. Would this affect the findings?

Response: The participants in this cohort were born at a time when childhood obesity was

uncommon. Thus we would argue that the term early-onset obesity for obesity at 20 years is appropriate for this group. However, we agree with the reviewer that such a definition may not be appropriate in younger cohorts experiencing childhood in an obesogenic environment. We have added this text to page 11 to highlight this: "Our measure of early-onset obesity is based on obesity at a mean age of 20 years; whilst we feel that this is an appropriate definition of early-onset obesity in this population, who grew up at a time when childhood obesity was very rare, studies in younger cohorts may wish to consider earlier ages to define 'early' obesity." We have discussed the limitation of not having BMI data before age 7 on pages 10 and 11.

9. These study questions have been focused upon in the Helsinki Birth Cohort Study – e.g. Salonen MK, Kajantie E, Osmond C, Forsén T, Ylihärsilä H, Paile-Hyvärinen M, Barker DJ, Eriksson JG. Role of childhood growth on the risk of metabolic syndrome in obese men and women. *Diabetes Metab.* 2009 Apr;35(2):94-100 and Salonen MK, Kajantie E, Osmond C, Forsén T, Ylihärsilä H, Paile-Hyvärinen M, Barker DJ, Eriksson JG. Childhood growth and future risk of the metabolic syndrome in normal-weight men and women. *Diabetes Metab.* 2009 Apr;35(2):143-50 especially the first paper focuses upon issues very similar to the present study and should be discussed.

Response: We thank the reviewer for this suggestion and have added the first reference to the manuscript on pages 4 ("Analyses of 499 obese adults in Helsinki, Finland, found that those who had the metabolic syndrome were similar in birth size compared with obese people who did not have metabolic syndrome, but were lighter and thinner between 2 and 11 years of age") and 10 ("The earliest available childhood BMI measures in our cohort are from age 7; thus it is possible that BMI changes at earlier stages could be associated with being metabolically healthy, and we are thus unable to make direct comparisons between our results and those of the studies looking at birth size and weight gain in early childhood in relation to MHO").

10. The age range being studied is extremely wide from 22 to 75 years, can this fully be taken into account by the statistical adjustments?

Response: In response to the reviewer's suggestion, we checked the effect of allowing for non-linearity in the association between age and the cardiometabolic risk factors by including age-squared and age-cubed in the models; this did not affect any of the results and we are therefore confident that adjusting for age alone has sufficiently taken into account the age differences amongst participants.

11. The problems associated with using BMI as a measure of obesity should be discussed in more detail.

Response: In response to the reviewer's suggestion, we have added the following text to page 11: "Our measures of adiposity are based on BMI, which is an imperfect measure of fatness, particularly in children. However, even in childhood, BMI has been shown to have similar associations with cardiovascular risk factors as directly-determined fat mass"

12. How representative is the study population participating in the follow-up to the whole original population?

Response: we thank the reviewer for highlighting this omission and have now added the following text to page 5: "Previous analyses of this cohort have shown that lower BMI and higher intelligence, educational level, social class and age were associated with a greater probability of attending follow-up clinics"

Minor issues:

1. Is this a prospective study or would it be more appropriate to call it longitudinal

Response: we agree it is longitudinal and have amended the manuscript title and abstract accordingly

2. Page 2, l 37/38: mean – should be men

Response: we have now corrected this error

Reviewer Name Carrie Durward

In the abstract, line 38 "mean" should be "men".

Response: we have now corrected this error

In some places (abstract, introduction) MHO is defined as "metabolically healthy, but obese", but other places (article summary, tables) it is "metabolically healthy, obese".

Response: we have now corrected this error and referred to MHO as "metabolically healthy, but obese" throughout